# Orbital perspective on high-harmonic generation from solids

Álvaro Jiménez-Galán[1,2,3], Chandler Bossaer[1,4], Guilmot Ernotte[1], Andrew M. Parks [4], Rui E. F. Silva[3], David M. Villeneuve [1], André Staudte[1], Thomas Brabec[4], Adina Luican-Mayer [4] & Giulio Vampa [1] ✉

High-harmonic generation in solids allows probing and controlling electron dynamics in crystals on few femtosecond timescales, paving the way to lightwave electronics. In the spatial domain, recent advances in the real-space interpretation of high-harmonic emission in solids allows imaging the field-free, static, potential of the valence electrons with picometer resolution. The combination of such extreme spatial and temporal resolutions to measure and control strong-field dynamics in solids at the atomic scale is poised to unlock a new frontier of lightwave electronics. Here, we report a strong intensity-dependent anisotropy in the high-harmonic generation from $ReS_2$ that we attribute to angle-dependent interference of currents from the different atoms in the unit cell. Furthermore, we demonstrate how the laser parameters control the relative contribution of these atoms to the high-harmonic emission. Our findings provide an unprecedented atomic perspective on strong-field dynamics in crystals, revealing key factors to consider in the route towards developing efficient harmonic emitters.

The foundational concept underpinning attosecond physics, and high-harmonic generation in gas-phase atoms and molecules in particular, is the energetic recollision of an electron ionized and accelerated by a strong laser field with the parent ion[1-4]. This dynamic real-space framework is instrumental to link the characteristics of the emitted harmonic radiation (amplitudes, phases and polarization) to sub-laser-cycle dynamics of atomic and molecular orbitals[5-9]. In solids, high-harmonic generation (HHG) is understood using a similar framework, albeit exchanging the real-space perspective for one in reciprocal space, where electron-hole pairs accelerate and recombine across energy bands in the Brillouin zone of the crystal[10-13]. This reciprocal-space approach has been paramount in virtually all investigations of solid-state high-harmonics: from revealing the role of electron-hole recollisions in the emission process[14], to reconstructing the band structure of a ZnO crystal[15], to explaining the multiple plateaus observed in the HHG spectrum[16] and to map regions of crystal momenta where the electron-hole velocity vanishes[17], among others[18-24].

Despite the success of the reciprocal-space picture, a real-space approach offers a more intuitive framework, in particular in complex materials with many narrowly spaced and overlapping bands. The advantages of using a real-space perspective to understand HHG from solids are quickly starting to become apparent[25-27], for example, in interpreting spatially-displaced electron-hole recollision processes[27,28] or as means to directly reconstruct the field-free (static) valence electron potential at the picometer scale[29]. The possibility to link features of the high-harmonic spectrum to dynamics occurring at specific orbitals in the lattice remains, however, largely unexplored. Here, we demonstrate this possibility through angle-resolved measurements of HHG in $ReS_2$. We measure a strong, intensity-dependent anisotropy of the HHG emission and trace it back to the interplay between the currents generated by each individual atom in the unit cell. Simulating the laser-matter interaction using a basis constructed from maximally-localized Wannier orbitals, we show that by changing the laser parameters (intensity and polarization), one can activate or suppress the

[1]Joint Attosecond Science Laboratory, National Research Council of Canada and University of Ottawa, Ottawa, ON K1A 0R6, Canada. [2]Max-Born-Institute, Max-Born Strasse 2A, D-12489 Berlin, Germany. [3]Instituto de Ciencia de Materiales de Madrid (ICMM), Consejo Superior de Investigaciones Científicas (CSIC), Sor Juana Inés de la Cruz 3, 28049 Madrid, Spain. [4]Department of Physics, University of Ottawa, Ottawa, ON K1N 6N5, Canada. ✉e-mail: gvampa@uottawa.ca

contribution of specific atoms to the HHG emission and interfere the atomic currents differently, increasing or decreasing the high-harmonic emission efficiency.

## Results

ReS$_2$ is a layered semiconductor that crystallizes in a distorted octahedral (T) phase[30–34]. Figure 1a illustrates the unit cell of the monolayer, formed by 4 rhenium atoms and 8 sulfur atoms. The 4 Re clusters are linked in a chain oriented along $\theta = 120°$ (see panel a). While the anisotropy of the crystal structure is clear (the crystal symmetry group is *P-1*), the band structure is similar along different angles and is very dense (see Fig. 1b and Supplementary Note 1), and with a density of states near the Fermi energy significantly higher than other prototypical materials used in HHG spectroscopy, such as MgO or ZnO[35]. Going from the monolayer limit to bulk, these features remain, and the band structure changes only slightly[30,31]. In such dense band diagram, associating an individual harmonic with reciprocal space trajectories of charge carriers in a particular set of bands, according to the reciprocal-space method, is hardly straightforward (see circular markers in Fig. 1b–d), and is unlikely to provide much insight into the carrier dynamics. On the other hand, the small bandwidth indicates that the electrons are very localized on the individual atoms of the lattice, making it ideally suited for a real-space or orbital-based framework.

The first question we want to address is if high harmonics generated from ReS$_2$ reflect the strong anisotropy apparent in real space or rather the weak angular dependence of its band structure. We generate high harmonics from bulk ReS$_2$ with a linearly-polarized mid-infrared pulse with a duration of 80 fs and a center wavelength of 3.5 μm (see Methods). Figure 2a shows the high-harmonic spectrum measured for a laser intensity of 0.64 TW/cm², measured in air, and polarization along $\theta = 120°$ (see inset). We observe odd harmonics extending up to the 13$^\text{th}$ order, while even harmonics are absent as expected from the inversion symmetry of ReS$_2$. We measure the orientation dependence of the harmonics by rotating the polarization of the linear pulse with respect to the crystal. The results, shown in Fig. 2b–d, display a clear anisotropy for all harmonic orders. Furthermore, the anisotropy depends strongly on the laser intensity.

In order to understand the origin of this anisotropy, we perform time-dependent simulations in a basis constructed from 44 maximally-localized Wannier orbitals (see Methods for details)[36]. The similarity between the monolayer and bulk forms in the case of ReS$_2$[30,31], allows us to reduce the computational complexity and simulate the monolayer system. The orientation dependence of H9 and H11 obtained from the numerical simulations is shown in Fig. 3a, b. While the uncertainty of the experimental intensities and the differences

between the monolayer and bulk forms do not allow for a quantitative experiment-theory comparison (e.g., of the exact position of the harmonic maxima), our simulations clearly display the strong intensity-dependent anisotropy observed in the experiment. As a result, the simulations in monolayer ReS$_2$ can provide valuable insight for the origin of this effect.

The high-harmonic spectrum is given by the Fourier components of the time-dependent current that is generated by the laser-induced oscillating dipole of the medium (see Methods),

$$I(\omega) = \sum_\alpha \left| \mathcal{F}[J_\alpha(t)](\omega) \right|^2, \tag{1}$$

where $J_\alpha(t)$ is the total current along direction $\alpha = (\|, \perp)$, corresponding to the components parallel and perpendicular to the electric field of the driving laser, respectively. The total current can be expressed as a sum of currents from all the orbitals in the lattice, $J_\alpha(t) = \sum_n^{N_\text{orb}} J_{n,\alpha}^{(W)}(t)$, where $J_{n,\alpha}^{(W)}(t)$ represents the contribution to the total current of the changing population of orbital $n$ and its coherence with all other orbitals (see Methods). The superscript (W) indicates that such orbital currents are defined in the Wannier gauge and, even if they are not observable, provide a unique real-space perspective into the HHG process. Expressed in terms of the individual orbital currents, the high-harmonic spectrum is

$$\begin{aligned}
I_\alpha(\omega) &= \left| \sum_n^{N_\text{orb}} \mathcal{F}[J_{n,\alpha}^{(W)}(t)](\omega) \right|^2 \\
&= \sum_n^{N_\text{orb}} \left[ |A_{n,\alpha}(\omega)|^2 + |A_{n,\alpha}(\omega)| \sum_{m \neq n} |A_{m,\alpha}(\omega)| \cos(\varphi_{m,\alpha}(\omega) - \varphi_{n,\alpha}(\omega)) \right],
\end{aligned} \tag{2}$$

where $A_{n,\alpha}$ and $\varphi_{n,\alpha}$ are, respectively, the spectral amplitude and phase of the current of orbital $n$ along direction $\alpha$.

Equation (2) allows us to distinguish features that arise from the interference of different orbital currents. The incoherent sum of the individual currents, $I_\alpha^\text{incoh}(\omega) = \sum_n^{N_\text{orb}} \left| \mathcal{F}[J_{n,\alpha}^{(W)}(t)](\omega) \right|^2$, will be absent of such interference. In Fig. 3c we compare the angle-dependent harmonic yield of H11 for $I_\alpha^\text{incoh}$ (solid lines) and the observable signal $I_\alpha$ (faint dashed lines). A similar analysis for H9 is made in Supplementary Note 2. The angular variation is stronger for $I_\alpha$, with near-complete suppression of various secondary maxima that are present in $I_\alpha^\text{incoh}$ (most notably near 60°), strongly modifying the orientation

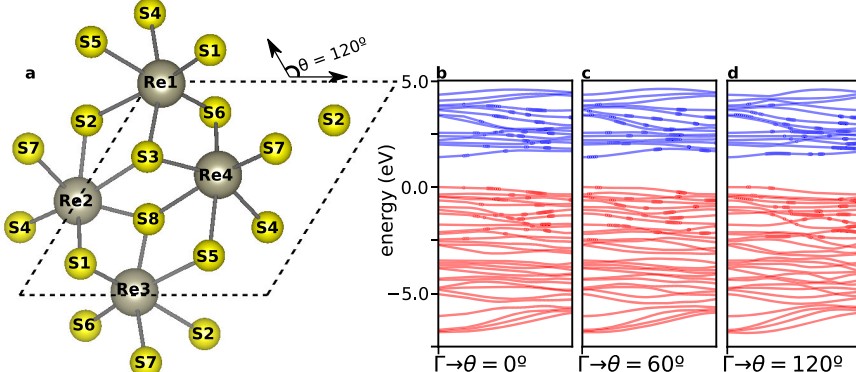

**Fig. 1 | Monolayer ReS$_2$. a** Crystal structure, composed of 4 rhenium atoms and 8 sulfur atoms in a distorted octohedral structure. The unit cell is delimited by the parallelogram. **b–d** Band structure of the monolayer along (a) $\theta = 0°$, (b) $\theta = 60°$ and (c) $\theta = 120°$ (see panel a for definition of $\theta$). Circular markers across the bands in panels **b–d** highlight vertical transitions resonant with the 11th harmonic (H11). Monolayer and bulk (see, e.g., references[30,31]) forms of ReS$_2$ are both inversion symmetric and display a very similar electronic band structure, with a nearly identical direct band gap of 1.4 eV at the Γ point[30,31].

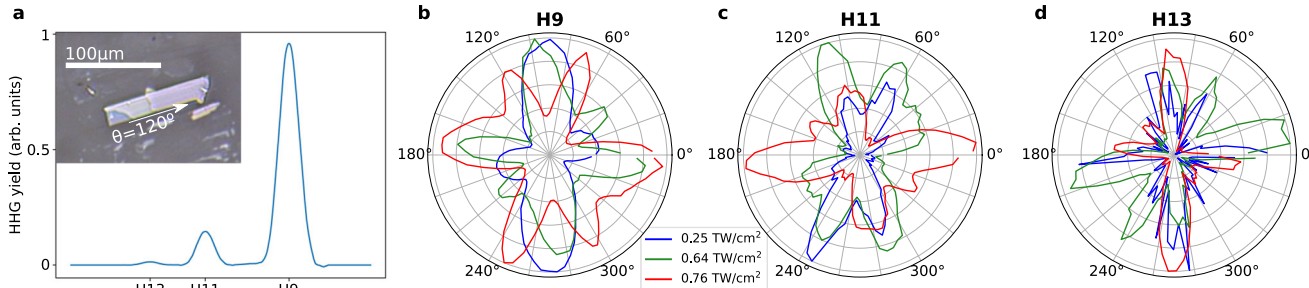

**Fig. 2 | Measured orientation-dependent HHG from ReS₂. a** High-harmonic spectrum for a laser intensity of 0.64 TW/cm² and polarization along $\theta = 120°$ (parallel to the rhenium chains). The inset shows an optical micrograph of the bulk ReS₂ flake using a CMOS camera and white-light illumination, with the longest edge corresponding to the rhenium chains. Orientation dependence of (**b**) H9, (**c**) H11 and (**d**) H13 for three different intensities: 0.25 TW/cm² (blue), 0.64 TW/cm² (green) and 0.76 TW/cm² (red). We note that in the experiment, the intensity is measured in air before impinging on the material.

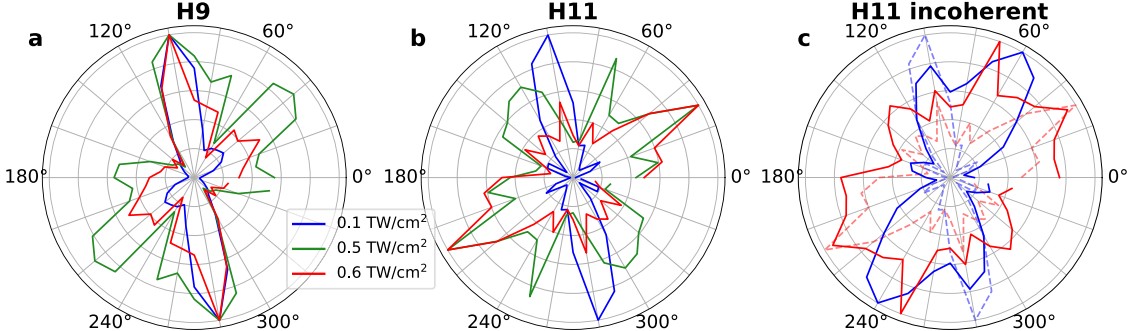

**Fig. 3 | Calculated orientation-dependent HHG from ReS₂.** Full calculation of harmonics (**a**) H9 and (**b**) H11 for different intensities: 0.1 TW/cm² (blue), 0.5 TW/cm² (green) and 0.6 TW/cm² (red). **c** Calculation neglecting the Fourier phase (solid lines) $\varphi_n$ of the orbital current for H11 for 0.1 TW/cm² (blue) and 0.6 TW/cm² (red). For comparison, the full calculation curves of panel **b** are shown in **c** with dashed, faint lines. We note that in the simulations, the intensity refers to that inside the material.

dependence. Thus, orbital phase interference is an important factor determining the orientation dependence.

Since the electrons are well localized on each atomic site (see Supplementary Data 1), we can group together the currents of the $m$ orbitals belonging to the same atom $A$ into an atomic current, $J_{A,\alpha}^{(W)}(t) = \sum_m J_{m,\alpha}^{(W)}(t)$. Furthermore, due to the inversion symmetry of ReS₂, each atom is related to one other by an inversion operation, for example, Re₁ and Re₃ or S₁ and S₆ (see Fig. 1a). Both of the atoms in the pair give rise to the same Fourier amplitudes and phases, so that the total harmonic spectrum in Eq. (2) reduces to the sum of the Fourier amplitudes and phases of six atomic (inversion-related) pairs.

Figure 4 a, b show the Fourier amplitudes and phases of the six atomic pairs, indicated with different colors, for H11 along $\alpha = \parallel$ and for two intensities: 0.1 TW/cm² and 0.6 TW/cm². At both low and high intensities (Fig. 4a, b respectively) emission is spread over a wide range of phases at any given angle. For the lowest intensity (Fig. 4a), every atomic pair contributes a similar amplitude to the emission near $\theta = 40–60°$, but their phases are spread equally over $\pi$ rad, thus leading to the near-perfect destructive interference seen in Fig. 3b (blue curve) at these angles. On the other hand, for angles close to $\theta = 100°$, the Fourier phases from the different atomic sites are similar, leading to constructive interference in Eq. (2) and therefore to the peak observed in Fig. 3b (blue curve). At $\theta = 100°$, the atomic pair Re₂-Re₄, which contributes the most to H11 at low intensity, is largely suppressed at high intensity (compare size of orange circle in Fig. 4a, b). This analysis shows that atoms that do not contribute to the generation of a particular harmonic order for one driver intensity, can be activated for other intensities, and vice versa, suggesting that laser intensity could be used as a mechanism to control the relative weight of atomic orbitals in HHG. An analogous analysis can be made for the rest of

harmonic orders, along both $\alpha = \parallel, \perp$ directions (see Supplementary Note 2), where we observe a larger spread of the Fourier phases for increasing harmonic orders. This leads to sharper changes in the angle-resolved spectrum for higher orders, as also seen in the experiment.

In conclusion, we identify how the nonlinear currents residing on each of the twelve atoms in the unit cell of a ReS₂ crystal are responsible for the strongly anisotropic and intensity-dependent emission of high-order harmonics. Our orbital analysis based on maximally localized Wannier functions reveals that each atomic contribution depends strongly on the polarization angle and intensity of the driving field, paving the way to characterizing and controlling electron dynamics at the picometer-scale in solids on sub-laser-cycle timescales. Moreover, we show that interference between atoms in the unit cell of a crystal is key to determine the macroscopic high-harmonic emission, a critical factor to consider in the route towards developing efficient harmonic emitters.

## Methods
### Experimental methods
ReS₂ flakes were mechanically exfoliated from an extracted section of a bulk sample using tape, then dispersed across the tape by folding over itself to reduce thickness and produce generally flat flakes. The ReS₂ crystal was transferred from the tape to a PDMS stamp, then transferred from the stamp to the substrate at 80°. The substrate consists of a two-side polished, $10 \times 10 \times 0.5$ mm, (100)-cut MgO single crystal that is cleaned with acetone and isopropanol. The PDMS stamp was peeled off to leave the bulk ReS₂ flakes on the MgO substrate.

The sample is imaged in-situ with a white-light source as well as the laser source, allowing sample areas of interest to be located and crystallographic orientation to be measured.

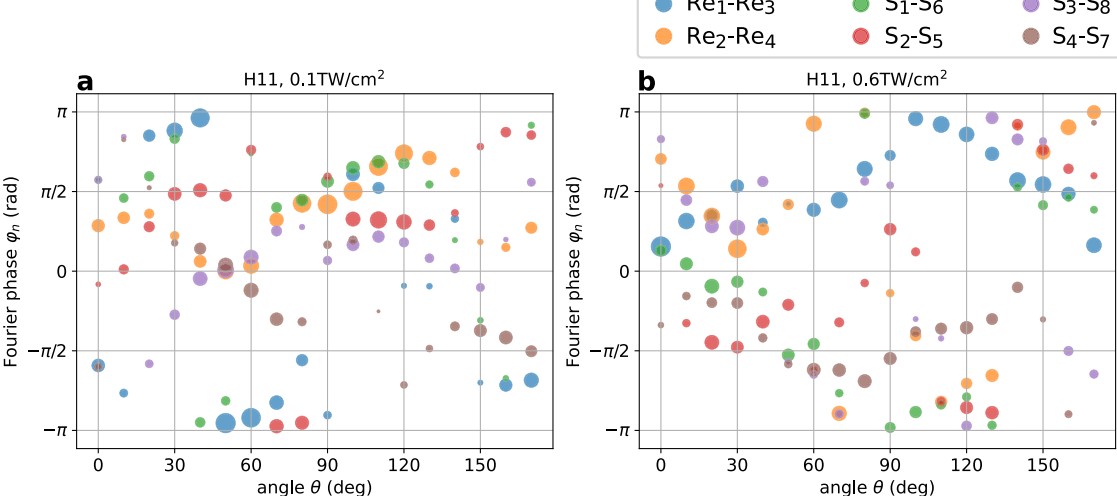

**Fig. 4 | Atomic contributions to harmonic emission in ReS₂.** The circles colors represent the six different atomic pairs, the size of each circle is proportional to the Fourier amplitude $|A_n|$ and the Fourier phase $\varphi_n$ is given in the vertical axis. The panels display the Fourier amplitudes $|A_n|$ and phases $\varphi_n$ of the six atomic pairs (inversion-symmetric partners) as a function of the laser polarization angle for H11. Two driver intensities are shown: **a** 0.1 TW/cm² and **b** 0.6 TW/cm². The results shown are for the harmonic polarization $\alpha$ that is parallel to the electric field.

The laser source consists of a Yb:KGW laser (LightConversion Carbide CB3) delivering 200fs pulses at a center wavelength of 1030 nm with a repetition rate of 100 kHz and average power of 80 W. A portion of this power (60 W) pumps a commercial optical parametric amplifier (LightConversion Orpheus-MIR), generating mid-infrared pulses at a wavelength of 3.5 μm with a bandwidth that sustains a 60 fs pulse. An Ag off-axis parabolic mirror focuses the mid-infrared beam onto the sample, producing high harmonics. The size of the focus is estimated from the size of the damaged region of the flake after illuminating it at a high power and was measured to be $58 \pm 15$ μm. The intensity was then obtained from the focus size, pulse duration, repetition rate and measured power, yielding values compatible with those from theory within a range of 60%. The generated high harmonics are collected in transmission geometry and focused on the input slit of a Princeton Instruments IsoPlane spectrometer with an Al spherical mirror of 15 cm focal length. A half-wave plate is positioned between the parabolic mirror and the sample to rotate the linear laser polarization with respect to the crystal axis.

**Numerical methods**
The field-free Hamiltonian and dipole couplings of monolayer ReS₂ were calculated with the electronic structure code Quantum Espresso[37] on a Monkhorst–Pack (MP) grid of $12 \times 12 \times 1$ points using a norm-conserving Perdew–Burke–Ernzerhof (PBE) exchange correlation functional. The field-free Hamiltonian used in the time-dependent propagation was constructed by projecting the Bloch states onto a set of maximally-localized Wannier functions using the Wannier90 code[36]. In particular, we projected onto the $d$ orbitals of the four rhenium atoms and the $p$ orbitals of the six sulfur atoms, totaling 44 bands. The Hamiltonian in the basis constructed from Wannier functions was then propagated in the presence of the electric field using the density matrix formalism with the code described in ref. 26. The large size of the unit cell allowed us to obtain convergence with a modest MP grid of $50 \times 50$ $k$-points along the $b_1$ and $b_2$ reciprocal lattice vectors. The time step was set to 0.2 a.u. and the dephasing time was chosen to be $T_2 = 10$ fs. While the exact position of the peaks and the broadness of the angular features at fixed intensities depend on the choice of dephasing time, the intensity-dependent strong anisotropy observed is independent on the choice of $T_2$, as well as on small changes of the central wavelength (see Supplementary Note 3).

The time-dependent current along direction $\alpha$, used to extract the high harmonic spectrum, is defined as

$$J_\alpha(t) = -\frac{|e|}{N_k} \sum_{\mathbf{k}} \mathrm{Tr}\left[\hat{\mathbf{v}}_\alpha(\mathbf{k}) \cdot \rho(\mathbf{k},t)\right]. \quad (3)$$

Above, $e$ is the electron charge, $N_k$ is the number of crystal momenta included in the calculation, $\hat{\mathbf{v}}$ is the velocity operator, and $\rho$ is the density matrix. In the Wannier gauge, the density matrix $\rho^{(W)}$ contains the orbital populations and coherences in its diagonal and off-diagonal terms, respectively. In the Wannier gauge, we may define a (real) current from an individual orbital $n$ along direction $\alpha$ as

$$J_{n,\alpha}^{(W)}(t) = -\frac{|e|}{N_k} \mathrm{Re}\left\{ \sum_{\mathbf{k}} \sum_{m}^{N_{orb}} \left[\hat{v}_{nm,\alpha}^{(W)} \cdot \rho_{mn}^{(W)}\right] \right\}, \quad (4)$$

such that the sum of the currents from all orbitals equals the total current,

$$J_\alpha(t) = \sum_{n}^{N_{orb}} J_{n,\alpha}^{(W)}(t). \quad (5)$$

For clarity, we give an example for a two-orbital model, although we point out that our analysis is only relevant for multi-orbital crystals as the one presented in this work. In the two-orbital case,

$$J_{1,\alpha}(t) = \mathrm{Re}\{v_{11,\alpha}\rho_{11} + v_{12,\alpha}\rho_{21}\}$$
$$J_{2,\alpha}(t) = \mathrm{Re}\{v_{22,\alpha}\rho_{22} + v_{21,\alpha}\rho_{12}\}, \quad (6)$$

where the subscripts 1 and 2 identify the orbital and $\alpha = \parallel, \perp$ the direction of current emission. Since both the velocity and density matrices are hermitian, the current of an individual orbital is composed of a term associated to the population change of that orbital, plus exactly half of the contribution of the coherence between that orbital and the rest. Thus, this approach offers a way of quantifying the contribution of individual orbitals, and their interference, to the high-harmonic generation.

## Data availability
The data that support the plots within this paper and other findings of this study are available from the corresponding author upon request.

## Code availability

Three numerical codes were used in this work: Quantum Espresso, Wannier90 and Iweria. The first two are open-source and can be found, respectively, in https://www.quantum-espresso.org/and https://wannier.org/. Iweria is an in-house code described in ref. 26, and parts of it relevant to reproducing the results of this work can be made available from the corresponding author upon request.

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

## Acknowledgements

We thank David Crane, Ryan Kroeker and Andrei Naumov for continued technical support. A.J.G. acknowledges support from the European Union's Horizon 2020 research and innovation program under the Marie Skłodowska-Curie Grant Agreement (101028938), from Comunidad de Madrid through TALENTO Grant 2022-T1/IND-24102 and from the Joint Center for Extreme Photonics. G.V. acknowledges support from the Joint Center for Extreme Photonics and the National Research Council's Quantum Sensing program. D.V. acknowledges support from the Joint Center for Extreme Photonics. R.E.F.S. acknowledges support from the fellowship LCF/BQ/PR21/11840008 from "La Caixa" Foundation (ID 100010434).

## Author contributions

C.B. and G.E. performed the experiments; A.J.G. performed the numerical analysis and calculations; A.P. and R.E.F.S. performed ancillary calculations; R.E.F.S. developed the numerical code; D.M.V., A.S., T.B., A.L.M. and G.V. supervised the work. A.J.G. and G.V. wrote the manuscript with contributions from all co-authors.

## Competing interests

The authors declare no competing interests.
