## [Peer Review File · Nature Communications]

REVIEWER COMMENTS

Reviewer #2 (Remarks to the Author):

The authors of the manuscript titled 'Orbital perspective on high-harmonic generation from solids' describe experimental results supported by simulations on high-harmonic generation from ReS₂ crystals where they found interesting features that can be explained in the real-space perspective of strong-field solid-state physics. In their experiments, where they studied angle-dependent intensity of the generated radiation, they observed an intensity-dependent change in the strength of the generated harmonics which changes as a function of the applied driving field strength (or intensity). This they attribute to the destructive and constructive interference of radiation originating from currents in different atoms of the unit cell. The methods (both experimental and theoretical) the authors use fit to the needs of the study, are top level, and used appropriately based on the given details. The study is very elaborate, especially with the supplementary material, showing that the authors have examined the findings in big detail. At the same time, I find their claims sometimes not matched with or supported by the results, so I suggest revision of their article according to the following before acceptance can be suggested:

- The abstract of the work ends with the statement 'suggest that crystals with a large number of atoms in the unit cell are not necessarily more efficient harmonic emitters than those with fewer atoms'. It is not clear for me, even after reading through the work, that why someone would assume that a crystal with higher number of atoms in the unit cell be a more efficient source of high-harmonics than one with lower number. I can imagine that like in the case of harmonic from gases, a larger number of emitters from the interaction volume potentially gives higher flux (but their phase matching is an important factor, in some sense similar aspect to the interference studied in this work). But a larger number of atoms in the unit cell does not necessarily mean a larger number of emitters in the interaction volume, because, e.g. the size of the unit cell can also be very different. The authors should revisit this statement and give a clearer explanation.

- On page 5 it is highlighted ('while the uncertainty of the experimental intensities') that there is an uncertainty of laser intensity. This seems to be an important factor in experiment vs. simulation comparability, so I find it important to quantify the uncertainty of this value with an error margin.

- On the same page, it is explicitly mentioned that 'e.g., of the exact position of the harmonic maxima' are not quantitatively comparable. While this is absolutely true and understandable, I would still expect some more quantitative comparison possible. For example, in the experimental figures (Fig 2b-d) I see a tendency that higher harmonic signal (H9 -> vs H13) has more sharp anisotropy features than lower orders (sharper 'intensity blobs as a function of angle'). This feature does not seem to reappear in the simulations (Fig. 3). Also, experiments show a larger number of angles with constructive interference (larger harmonic signal) compared to simulations. Are these observations reproducible? If yes, they are of physical origin, and I would expect them to be explainable by the simulations.

- The explanation of the observations in the simulations are very convincing (especially with the data in the supplementary note 2), and clearly shows that the explanation found is what lies behind the

observations. The weak point I see is the connection with the experiment or applicability. While there is no reason to doubt that the experimental observations are connected to simulations results and share physical origin, the uncertainties indicated above makes the conclusion 'paving the way to characterizing and controlling electron dynamics at the picometer-scale in solids on sub-laser-cycle timescales' a bit questionable. With the uncertainties in the intensity, is it really a control? For this it would be important to compare the error margin of experimental intensity with the anisotropy change expected in that intensity range from simulations.

Two additional minor remarks:

- In fig. 1(a) there seems to be an unnecessary S2 atom
- on page 6 'The subscript (W) indicates', W is a superscript actually

Reviewer #3 (Remarks to the Author):

This work takes a real-space approach to HHG in solids by analyzing the contributions to the HHG spectrum of individual atoms within the unit cell. The authors find that the HHG spectrum is not just determined by the band structure, but also by the interference between HHG signals coming from different atoms within the unit cell. This interference leads to a strong angular dependence of HHG, even though the band structure of a layered ReS₂ semiconductor is similar in different directions (suggesting HHG spectrum should not vary strongly with polarization direction of the laser).

The work is likely to be of broad interest to the community working on HHG in solids. I have a few concerns that should be addressed prior to publication.

1. One drawback is a lack of even a quantitative agreement between theory and experiment. I realize this is not likely to be remedied in the revision, but I would like the authors to go deeper into the possible cause for this lack of agreement.
2. It would be nice to link back to k-space, since ultimately, the two descriptions (in real vs. k-space) are equivalent. Hence, any interference in real space should have a corresponding analogue in k-space.
3. How does the finding that HHG is not determined just by the band structure square with the prior findings by some of the same authors that reconstructs band structure based on HHG emission (see Vampa et al PRL 193603, 2015)? Are there conditions the authors can give for when it is possible to apply the method in PRL 193603 to reconstruct the band structure and when it will fail?

4. The work is using 3.5 microns on a material with direct bandgap of 1.4 eV. Hence the direct bandgap roughly corresponds to 4 photons. This is lower than the typical HHG process where the bandgap is normally roughly an order of magnitude bigger than the photon energy. Have the authors looked at the scaling of HHG with intensity and whether this scaling suggests a perturbative process?

What is the Keldysh parameter of this experiment?

5. Related to 4. above, what is the mechanism of HHG? In particular, is it interband or intraband? I realize this requires switching to the k-space picture, but a better understanding of the underlying mechanism would improve the manuscript.

Referee 2

The methods (both experimental and theoretical) the authors use fit to the needs of the study, are top level, and used appropriately based on the given details. The study is very elaborate, especially with the supplementary material, showing that the authors have examined the findings in big detail. At the same time, I find their claims sometimes not matched with or supported by the results, so I suggest revision of their article according to the following before acceptance can be suggested:

We thank the referee for the positive evaluation of our work and for the critical remarks regarding some of the claims and the experiment-theory comparison, which we address below.

1. The abstract of the work ends with the statement 'suggest that crystals with a large number of atoms in the unit cell are not necessarily more efficient harmonic emitters than those with fewer atoms'. It is not clear for me, even after reading through the work, that why someone would assume that a crystal with higher number of atoms in the unit cell be a more efficient source of high-harmonics than one with lower number. I can imagine that like in the case of harmonic from gases, a larger number of emitters from the interaction volume potentially gives higher flux (but their phase matching is an important factor, in some sense similar aspect to the interference studied in this work). But a larger number of atoms in the unit cell does not necessarily mean a larger number of emitters in the interaction volume, because, e.g. the size of the unit cell can also be very different. The authors should revisit this statement and give a clearer explanation.

We thank the referee for this comment, which made us realize that indeed this statement was poorly phrased, i.e., we implicitly assumed a fixed unit cell. Intuitively, one would expect that the parameter that determines a higher harmonic signal is roughly the density of atoms in the interaction volume. Yet, even in this case, one could also object that it would greatly depend on the atomic species. Since this statement is unprecise and does not affect the phenomena or conclusions described in the work we have rephrased as

"Our findings provide an unprecedented atomic perspective on strong-field dynamics in crystals, revealing key factors to consider in the route towards developing efficient harmonic emitters"

2. On page 5 it is highlighted ('while the uncertainty of the experimental intensities') that there is an uncertainty of laser intensity. This seems to be an important factor in experiment vs. simulation comparability, so I find it important to quantify the uncertainty of this value with an error margin.

As we see from the conclusions of this work, the angle-dependent harmonic

yield is very dependent on the driver intensity, which is hard to characterize with certainty in HHG experiments. On the one hand, this is attributable to the difficulty in estimating the size of the focus on target, as well as the exact duration of the pulse. On the other, the experimental intensity is measured in air, while the theoretical intensity refers to the intensity in matter, and is additionally assumed to remain unchanged since we neglect propagation effects inside the material. But, most importantly, limitations in the current theory make it extremely challenging to obtain a quantitative agreement with experiment. As we argue in our reply to the next question, even small discrepancies in the energy scales of the problem (e.g., uncertainties in band separations of $\simeq 20\text{meV}$) or different timescales of dephasing can produce a shift in the angle-resolved HHG peaks. These theoretical uncertainties are inevitable. Going beyond the theory that we use here is extremely challenging in this problem, where one needs to describe the strong-light matter interaction in a crystal with 44 bands. To the best of our knowledge, our present calculations are state-of-the-art.

Hence, a direct quantitative comparison between experiment and theory is out of the scope of the current work. This is why we have focused instead on comparing the *trend of the harmonic yield with increasing intensity*. We have made sure that *all of our observations and conclusions are independent on such theoretical limitations*, as we note in the reply to the next question.

For completeness, below we explain how we estimated the experimental intensity. The focus size was estimated from the size of the damaged spot on the ReS₂ flake after illuminating it with high power. We estimated the spot size to be $58 \pm 15 \mu\text{m}$. The pulse duration was estimated from the measured spectral bandwidth at installation, which yielded a pulse compatible with 60 fs duration. The experimental intensities reported were then calculated from the measured spot size, pulse duration, repetition rate of the laser and measured power.

Additionally, we performed theoretical simulations for other intensities in order to help us corroborate the estimated experimental intensities and their uncertainties, shown in Fig. 1. While it should be clear for the reasons above that we shouldn't look for quantitative agreement, it is clear that there are some common features and trends that allow us to check that the estimated experimental intensities are compatible with those from theory within a reasonable 60% offset. For example, one can distinguish between a one/two-peak structure at low intensities and a multi-peak structure at higher intensities. The left column of Fig. 1 corresponds to the lowest experimental and theoretical intensity reported in the manuscript (blue curves in Fig.2 and 3 of the manuscript), which differ by 60% of the experimental value. The central column corresponds to the experimental intensity shown by the green curve in Fig.2 of the main text, which was estimated to be $I_{\text{exp}} = 0.64 \text{ TW/cm}^2$. In order to match the angle-resolved HHG spectrum as best as possible, we averaged the spectrum of three theoretical intensities: $I_{\text{th}} = 0.3 \text{ TW/cm}^2 + 0.4 \text{ TW/cm}^2 + 0.5 \text{ TW/cm}^2$ which are close to the experimental intensity by an offset of less than 50%.

Finally, the highest intensity was estimated to be $I_{\text{exp}} = 1.15 \text{ TW/cm}^2$ (not shown in the manuscript since it is very close to the damage threshold). The theoretical intensity that best matched this spectrum is $I_{\text{th}} = 0.9 \text{ TW/cm}^2$, which differs by 22% of the experimental intensity. We have now added to the Methods section how the intensity was estimated:

“A portion of this power (60W) pumps a commercial optical parametric amplifier (LightConversion Orpheus-MIR), generating mid-infrared pulses at a wavelength of $3.5\mu\text{m}$ with a bandwidth that sustains a 60fs pulse. An Ag off-axis parabolic mirror focuses the mid-infrared beam onto the sample, producing high harmonics. The size of the focus is estimated from the size of the damaged region of the flake after illuminating it at a high power and was measured to be $58 \pm 15 \mu\text{m}$. The intensity was then obtained from the focus size, pulse duration, repetition rate and measured power, yielding values compatible with those from theory within a range of 60%.”

Figure 1: Comparison of angle-resolved HHG between theory and experiment for various intensities.

3. On the same page, it is explicitly mentioned that 'e.g., of the exact position of the harmonic maxima' are not quantitatively comparable. While this is absolutely true and understandable, I would still expect some more quantitative comparison possible. For example, in the experimental figures (Fig 2b-d) I see a tendency that higher harmonic signal (H9 -> vs H13) has more sharp anisotropy features than lower orders (sharper 'intensity blobs as a function of angle'). This feature does not seem to reappear in the simulations (Fig. 3). Also, experiments show a larger number of angles with constructive interference

(larger harmonic signal) compared to simulations. Are these observations reproducible? If yes, they are of physical origin, and I would expect them to be explainable by the simulations.

We thank the referee for pointing this out. We do see sharper anisotropy features in higher orders as compared to lower orders in both theory and experiment. Fig. 2 of this reply shows the theoretical angle-dependent harmonic spectrum for harmonics 3 to 15 for the three different intensities, showing sharper anisotropy features as a function of increasing harmonic order. In Supplementary Figures 5-8 we show the orbital phase vs angle plots, which show more scattered dots (leading to sharper anisotropic features) for increasing harmonic order. We have now incorporated the angle-resolved spectrum of H3-H15 in the Supplementary, and added the following sentence in the main text:

“An analogous analysis can be made for the rest of harmonic orders, along both $\alpha = \parallel, \perp$ directions (see Supplementary Note 2), where we observe a larger spread of the Fourier phases for increasing harmonic orders. This leads to sharper changes in the angle-resolved spectrum for higher orders, as also seen in the experiment.”

Figure 2: HHG anisotropy as a function of increasing harmonic order.

The second observation made by the referee – that the experiments show more angles with larger harmonic signal (or broader features) – is indeed not so clearly reproduced by the calculations. One of the main reasons is the way in which dephasing is incorporated into the theoretical description. As it is standard, we include dephasing as a phenomenological parameter T_2 which is constant throughout the Brillouin zone. Changing T_2 directly affects the coherences between orbitals. In this work, we have used a dephasing parameter $T_2 = 10$ fs, which corresponds approximately to one optical cycle, but we have

made sure not to make any claims which are dependent on this free parameter choice. This is also a reason why we did not seek a quantitative comparison with specific features in the experiment. Instead, we focus on the observation that the anisotropy is large (reflecting much more richness than that expected by its low crystal symmetry), and strongly intensity dependent. We have made sure that these observations are independent on the choice of T_2 .

To show the impact of the dephasing parameter, we have made calculations also for $T_2 = 2$ fs and $T_2 = 5$ fs. In Fig. 3, one can see that for short dephasing times, a larger number of angles show a larger harmonic signal as compared to longer dephasing times. In regards to this, we have made the following addition to the "Numerical methods" part of the main text:

"While the exact position of the peaks and the broadness of the angular features at fixed intensities depend on the choice of dephasing time, the intensity-dependent strong anisotropy observed is independent on the choice of T_2 ."

Figure 3: Dependence of dephasing on angle-resolved HHG in ReS_2 .

Finally, in Fig. 4 we also show how the angle-dependent HHG yield also changes for a detuning of the laser central wavelength ($3.5 \mu\text{m}$) of $\simeq 6\%$. We can see that some of the peaks appear shifted. Due to computational limitations, we have simulated a monolayer of ReS_2 , while the experiment is performed in bulk. While the band structure is similar in both systems [S. Tongay et al., Nat. Comm. **5** 3252 (2014); M. Gehlmann et al., Nano Letters **17** 5187 (2017)], Fig. 3 shows how even slight changes in the energy scales of the system can affect the angle-resolved HHG.

Figure 4: Dependence of laser frequency (energy scales) on angle-resolved HHG in ReS_2 .

4. *The explanation of the observations in the simulations are very convincing (especially with the data in the supplementary note 2), and clearly shows that the explanation found is what lies behind the observations. The weak point I see is the connection with the experiment or applicability. While there is no reason to doubt that the experimental observations are connected to simulations results and share physical origin, the uncertainties indicated above makes the conclusion 'paving the way to characterizing and controlling electron dynamics at the picometer-scale in solids on sub-laser-cycle timescales' a bit questionable. With the uncertainties in the intensity, is it really a control? For this it would be important to compare the error margin of experimental intensity with the anisotropy change expected in that intensity range from simulations.*

In the experiment, the control of electron dynamics in the lattice is represented by the appearance of new peaks (as well as by the cancelling of other peaks) in the angle-resolved harmonic spectrum, which theory links to changes in the amplitude of the non-linear currents associated to the individual lattice orbitals and their interferences. If we look at harmonic 9 for the lowest intensity (cf. blue curve of Fig.2a and Fig.3a of the manuscript), we see that both experiment and theory show a main peak at at around 110° (and $110^\circ+180^\circ$). As the intensity is increased from 0.25 TW/cm^2 to 0.64 TW/cm^2 in the experiment, new peaks appear (green curve). Similarly, in the theory new peaks appear as the intensity is increased. These extra peaks testify the control over the lattice orbitals, as shown by theory, and they appear in this example for an intensity change of $\sim 0.4 \text{ TW/cm}^2$, which is much larger than the experi-

mental uncertainty. One could of course have a more gradual level of control with a finer intensity scan, reaching down to the experimental uncertainty of the intensity. The fact that the precise peaks in experiment and theory do not exactly match is due to the reasons highlighted in the previous answers, which includes a driver intensity offset that is not precisely known. However, this does not invalidate that the control that we claim, exemplified in the experiment by the strong changes of the anisotropic distributions with intensity, is well within the experimental uncertainty.

Referee 3

The work is likely to be of broad interest to the community working on HHG in solids.

We thank the referee for the positive evaluation of our work.

1. One drawback is a lack of even a quantitative agreement between theory and experiment. I realize this is not likely to be remedied in the revision, but I would like the authors to go deeper into the possible cause for this lack of agreement.

We thank the referee for this remark, which we fully address below. There are several sources for the lack of quantitative agreement.

First, as we see from the conclusions of this work, the angle-dependent harmonic yield is very dependent on the driver intensity, which is hard to characterize with certainty in HHG experiments. On the one hand, this is attributable to the difficulty in estimating the size of the focus on target, as well as the exact duration of the pulse. On the other, the experimental intensity is measured in air, while the theoretical intensity refers to the intensity in matter, and is additionally assumed to remain unchanged since we neglect propagation effects inside the material. But, most importantly, limitations in the current theory make it extremely challenging to obtain a quantitative agreement with experiment. As we argue in the next paragraphs, even small discrepancies in the energy scales of the problem (e.g., uncertainties in band separations of $\simeq 20\text{meV}$) or different timescales of dephasing can produce a shift in the angle-resolved HHG peaks. These theoretical uncertainties are inevitable. Going beyond the theory that we use here is extremely challenging in this problem, where one needs to describe the strong-light matter interaction in a crystal with 44 bands. To the best of our knowledge, our present calculations are state-of-the-art.

Hence, a direct quantitative comparison between experiment and theory is out of the scope of the current work. This is why we have focused instead on comparing the *trend of the harmonic yield with increasing intensity*. We have made sure that *all of our observations and conclusions are independent on such theoretical limitations*, as we note in the next paragraphs.

For completeness, below we explain how we estimated the experimental intensity. The focus size was estimated from the size of the damaged spot on the ReS₂ flake after illuminating it with high power. We estimated the spot size to be $58 \pm 15 \mu\text{m}$. The pulse duration was estimated from the measured spectral bandwidth at installation, which yielded a pulse compatible with 60 fs duration. The experimental intensities reported were then calculated from the measured spot size, pulse duration, repetition rate of the laser and measured power.

Additionally, we performed theoretical simulations for other intensities in order to help us corroborate the estimated experimental intensities and their uncertainties, shown in Fig. 1. While it should be clear for the reasons above that we shouldn't look for quantitative agreement, it is clear that there are some common features and trends that allow us to check that the estimated experimental intensities are compatible with those from theory within a reasonable 60% offset. For example, one can distinguish between a one/two-peak structure at low intensities and a multi-peak structure at higher intensities. The left column of Fig. 1 corresponds to the lowest experimental and theoretical intensity reported in the manuscript (blue curves in Fig.2 and 3 of the manuscript), which differ by 60% of the experimental value. The central column corresponds to the experimental intensity shown by the green curve in Fig.2 of the main text, which was estimated to be $I_{\text{exp}} = 0.64 \text{ TW/cm}^2$. In order to match the angle-resolved HHG spectrum as best as possible, we averaged the spectrum of three theoretical intensities: $I_{\text{th}} = 0.3 \text{ TW/cm}^2 + 0.4 \text{ TW/cm}^2 + 0.5 \text{ TW/cm}^2$ which are close to the experimental intensity by an offset of less than 50%. Finally, the highest intensity was estimated to be $I_{\text{exp}} = 1.15 \text{ TW/cm}^2$ (not shown in the manuscript since it is very close to the damage threshold). The theoretical intensity that best matched this spectrum is $I_{\text{th}} = 0.9 \text{ TW/cm}^2$, which differs by 22% of the experimental intensity. We have now added to the Methods section how the intensity was estimated:

"A portion of this power (60W) pumps a commercial optical parametric amplifier (LightConversion Orpheus-MIR), generating mid-infrared pulses at a wavelength of $3.5\mu\text{m}$ with a bandwidth that sustains a 60fs pulse. An Ag off-axis parabolic mirror focuses the mid-infrared beam onto the sample, producing high harmonics. The size of the focus is estimated from the size of the damaged region of the flake after illuminating it at a high power and was measured to be $58 \pm 15 \mu\text{m}$. The intensity was then obtained from the focus size, pulse duration, repetition rate and measured power, yielding values compatible with those from theory within a range of 60%."

Figure 1: Comparison of angle-resolved HHG between theory and experiment for various intensities.

Regarding theory, one of the main limitations is the way in which dephasing is incorporated into the description of the light-matter interaction. As it is standard, we include dephasing as a phenomenological parameter T_2 which is constant throughout the Brillouin zone. Changing T_2 directly affects the coherences between orbitals. In this work, we have used a dephasing parameter $T_2 = 10$ fs, which corresponds approximately to one optical cycle, but we have made sure not to make any claims which are dependent on this free parameter choice. This is a reason why we did not seek a quantitative comparison with specific features in the experiment. Instead, we focus on the observation that the anisotropy is large (reflecting much more richness than that expected by its low crystal symmetry), and strongly intensity dependent. We made sure that these observations are independent on the choice of T_2 .

To show the impact of the dephasing parameter, we have made calculations also for $T_2 = 2$ fs and $T_2 = 5$ fs. In Fig. 2, one can see that for short dephasing times, a larger number of angles show a larger harmonic signal as compared to longer dephasing times. In regards to this, we have made the following addition to the "Numerical methods" part of the main text:

"While the exact position of the peaks and the number of angles showing a large harmonic signal at fixed intensities depend on the choice of dephasing time, the intensity-dependent strong anisotropy observed is independent on the choice of T_2 ."

Figure 2: Dependence of dephasing on angle-resolved HHG in ReS_2 .

Finally, in Fig. 3 we also show how the angle-dependent HHG yield also changes for a detuning of the laser central wavelength ($3.5 \mu\text{m}$) of $\simeq 6\%$. We can see that some of the peaks appear shifted. Due to computational limitations, we have simulated a monolayer of ReS_2 , while the experiment is performed in bulk. While the band structure is similar in both systems [S. Tongay et al., Nat. Comm. **5** 3252 (2014); M. Gehlmann et al., Nano Letters **17** 5187 (2017)], Fig. 3 shows how even slight changes in the energy scales of the system can affect the angle-resolved HHG. **We have incorporated the results for different dephasing times and different detunings of the central wavelength into the Supplementary.**

Figure 3: Dependence of laser frequency (energy scales) on angle-resolved HHG in ReS_2 .

2. It would be nice to link back to k -space, since ultimately, the two descriptions (in real vs. k -space) are equivalent. Hence, any interference in real space should have a corresponding analogue in k -space.

Indeed, this is an important point. Both k -space and real space pictures should of course give the same results. While the band structure looks isotropic, the dipole couplings are not, which ultimately leads to the HHG anisotropy. In Fig. 4 we show the k -dependent dipole couplings. Given the complexity of the bandstructure and of the dipole couplings, although reciprocal and real space are equivalent, the real space perspective that we extract in our work provides a much more intuitive framework to understand strong-field interactions in solids, especially in materials with a large number of “flat” bands. For such systems, one is unlikely to develop much intuition from a k -space approach.

Figure 4: Dipole couplings along the $X - \Gamma - Y$ direction between (a,d) lowest conduction band and five highest valence bands, (b,e) second lowest conduction band and five highest valence bands, (c,f) third lowest conduction band and five highest valence band. The panels (a-c) in the upper row are along the x ($\theta = 0^\circ$) direction. The panels (d-f) in the lower row are along the y ($\theta = 90^\circ$) direction.

3. How does the finding that HHG is not determined just by the band structure square with the prior findings by some of the same authors that reconstructs band structure based on HHG emission (see Vampa et al PRL 193603, 2015)? Are there conditions the authors can give for when it is possible to apply the method in PRL 193603 to reconstruct the band structure and when it will fail?

The reconstruction as highlighted in PRL 196303 is valid as long as there are two (or few) bands involved in the emission process, and that these bands are sufficiently wide (with a sizeable group velocity). This approximation is not valid in a system like ReS_2 .

4. The work is using 3.5 microns on a material with direct bandgap of 1.4 eV. Hence the direct bandgap roughly corresponds to 4 photons. This is lower than the typical HHG process where the bandgap is normally roughly an order of magnitude bigger than the photon energy. Have the authors looked at the scaling of HHG with intensity and whether this scaling suggests a perturbative process? What is the Keldysh parameter of this experiment?

We first want to note that our analysis and the framework we introduced does not depend on the regime (perturbative or tunnelling). In Fig. 5 we show the scaling of the angle-integrated experimental HHG yield with the driver intensity, with a power law to guide the eye. The scaling suggests that it is non-perturbative. Estimating the Keldysh parameter in a system such as ReS_2

is problematic since, on the one hand, there are too many bands close together that participate in the dynamics and, on the other, the bands have nearly flat dispersion, from which one cannot estimate a proper effective mass. This highlights once again the advantage of analyzing the dynamics with our orbital-based scheme.

Figure 5: Scaling of HHG yield as a function of the driver intensity for H9 (blue), H11 (green) and H13 (brown). Dashed black line is a power law to guide the eye.

5. Related to 4. above, what is the mechanism of HHG? In particular, is it interband or intraband? I realize this requires switching to the k -space picture, but a better understanding of the underlying mechanism would improve the manuscript.

In Fig. 6, 7 we show the harmonic spectrum separated in interband and intraband components. As can be seen, interband (red curve) dominates for all harmonic orders, for both low and high driver intensities, which reflects the fact that band structure is highly dense and flat.

Figure 6: Angle-resolved HHG separated in (blue) intra- and (red) inter-band contributions. The driver intensity is $I = 0.1 \text{ TW/cm}^2$.

Figure 7: Angle-resolved HHG separated in (blue) intra- and (red) inter-band contributions. The driver intensity is $I = 0.6 \text{ TW/cm}^2$.

REVIEWERS' COMMENTS

Reviewer #2 (Remarks to the Author):

The authors have adequately and convincingly addressed all reviewer concerns. I suggest acceptance of this work for publication in Nature Communications.

Reviewer #3 (Remarks to the Author):

I am largely satisfied with the authors' responses to my comments. I am still puzzled by the intensity dependence of HHG, shown in Figure 5 of the reply letter. Normally, one sees a power-law dependence at lower intensities and the flattening out of the curve at higher intensities. It does not look like the curves are following the I^n power law at lower intensities. This may be exacerbated by the experimental difficulties of calibrating the intensity, which the authors discuss at length in their response. However, I am happy to leave this to future investigation, as I don't think this question can be quickly resolved.

Referee 2

The authors have adequately and convincingly addressed all reviewer concerns. I suggest acceptance of this work for publication in Nature Communications.

We thank the referee for the positive evaluation of our work and our reply.

Referee 3

I am largely satisfied with the authors' responses to my comments. I am still puzzled by the intensity dependence of HHG, shown in Figure 5 of the reply letter. Normally, one sees a power-law dependence at lower intensities and the flattening out of the curve at higher intensities. It does not look like the curves are following the I^n power law at lower intensities. This may be exacerbated by the experimental difficulties of calibrating the intensity, which the authors discuss at length in their response. However, I am happy to leave this to future investigation, as I don't think this question can be quickly resolved.

We thank the referee for the positive evaluation of our work and our reply. Regarding the intensity dependence of HHG, we completely agree that it is common to see a power-law dependence at low intensities, which is especially true for low-order harmonics. Here, we believe it is not the case for two reasons. First, the lowest harmonic we measured was H9, and starting from an intensity which allowed enough counts to clearly see the harmonic order. Therefore, we believe that already for the intensity necessary to observe this harmonic order, the process is already non-perturbative. Second, given the strong intensity-dependent anisotropy in this material, the harmonic yield along a specific angle might even drop for some higher driver intensity as an effect of orbital-phase interference. In the plot that we attached in our previous reply, we computed the intensity dependence of the yield averaged over all angles. That said, we fully agree with the referee that intensity-dependent studies in such anisotropic materials is a highly interesting topic for future works, and we believe our orbital-based perspective will be a powerful tool to investigate it.